# Feasibility of a screening and prevention procedure for risks associated with dysphagia in older patients in geriatric units: the DYSPHAGING pilot study protocol

Olivier Durlach,[1] Stéphanie Tripoz-dit-Masson,[2] Nicolas Massé-Deragon,[1] Fabien Subtil,[3,4] Zeinabou Niasse-Sy,[1,5,6] Chloé Herledan,[7,8] Laure Guittard,[9,10] Karine Goldet,[2] Salima Merazga,[11] Margaux Chabert,[11] Anne Suel,[11] David Dayde,[12] Marion Merdinian,[1,13] Claire Falandry 🔵 [1,14,15]

**Correspondence to**
Dr Claire Falandry;
claire.falandry@chu-lyon.fr

## ABSTRACT

**Background** Dysphagia, particularly sarcopenic dysphagia, is frequent in frail older patients. Sarcopenic dysphagia is a swallowing disorder caused by sarcopenia, corresponding to a loss of muscle mass and strength. It frequently leads to inhalation and to the decrease of food intake, leading the patient to enter a vicious circle of chronic malnutrition and frailty. The awareness of the major health impacts of sarcopenic dysphagia is recent, explaining a low rate of screening in the population at risk. In this context, methods of prevention, evaluation and intervention of sarcopenic dysphagia adapted to the most at-risk population are necessary.

**Methods** The DYSPHAGING (dysphagia & aging) pilot study is a prospective, multicentre, non-comparative study aiming to estimate the feasibility of an intervention on allied health professionals using the DYSPHAGING educational sheet designed to implement a two-step procedure 'screen–prevent' to mitigate swallowing disorders related to sarcopenic dysphagia. After obtaining oral consent, patients are screened using Eating Assessment Tool-10 Score. In case of a score≥2, procedures including positional manoeuvres during mealtimes, food and texture adaptation should be implemented. The primary endpoint of the study is the feasibility of this two-step procedure (screening–prevention measures) in the first 3 days after patient's consent.

The study will include 102 patients, with an expected 10% rate of non-analysable patients. Participants will be recruited from acute geriatric wards, rehabilitation centres and long-term care units, with the hypothesis to reach a feasibility rate of 50% and reject a rate lower than 35%.

**Ethics and dissemination** The study protocol was approved according to French legislation (CPP Ile-de-France VII) on 15 February 2023. The results of the primary and secondary objectives will be published in peer-reviewed journals.

**Trial registration number** NCT05734586.

## STRENGTHS AND LIMITATIONS OF THIS STUDY

⇒ The DYSPHAGING study is a pilot study focusing on geriatric patients in different care sectors.
⇒ This study is based on a screening questionnaire recognised and used for the evaluation and follow-up of patients who benefit from rehabilitation and preventive measures of swallowing disorders complications.
⇒ The DYSPHAGING study is a prospective pilot study that aims to estimate the feasibility of this intervention.
⇒ Particular attention will be paid to the satisfaction of the nursing teams involved in the implementation of the questionnaire.

## INTRODUCTION

### Background and rationale

Sarcopenic dysphagia[1] is a swallowing disorder (or oropharyngeal dysphagia (OD)) resulting from the expression of sarcopenia, characterised by the loss of muscle mass and strength due to age and chronic diseases, in the oropharyngeal tract. This condition gives rise to critical complications related to inhalation risks[2,3] and exacerbates chronic undernutrition,[4] creating a detrimental cycle. Although recent awareness of the high prevalence of sarcopenic dysphagia and its severe consequences among older individuals with disabilities and hospitalised patients has grown, the screening within the affected population remains low and challenging, leading to suboptimal care.[5] In response, there is a pressing need for tailored prevention, assessment and intervention methods specifically designed for this vulnerable demographic.

To address this issue, the European Society for Swallowing Disorders and the European Union Geriatric Medicine Society have jointly developed a Dysphagia Working Group and published a white paper considering OD as a geriatric syndrome.[1] This position paper advocated for increased awareness of swallowing disorders, utilisation of screening scores, preventive measures, standardised diagnostics and implementation of targeted interventions.

In adherence to these recommendations, we have collaboratively developed a pedagogical tool, entitled DYSPHAGING form, within our multidisciplinary unit, following a comprehensive four-step approach: (1) screening, (2) protection, (3) diagnosis confirmation and (4) rehabilitation. The form was designed to allow, in routine care, a rapid screening and protection procedure. Using standardised questionnaires and a simple, schematic iconography, it is expected to be handled in routine by nurses, care assistants and even caregivers. As a first step, the DYSPHAGING pilot study was designed to evaluate the feasibility of this screening and protection in diverse geriatric wards (acute care, rehabilitation and long-term care units).

## METHODS AND ANALYSIS
### Objectives
#### Primary objective
The primary objective of DYSPHAGING pilot study is to assess the feasibility of implementing steps 1 and 2 of the DYSPHAGING form in hospital care units within 3 days after the patient's inclusion in the protocol.

#### Secondary objectives
Secondary objectives include: measurement of the percentage of eligible patients who refuse to participate in the study, characterisation of the target population (demographic and geriatric characteristics), quantification of non-implementation of protocol steps and reasons, description of factors associated with the risk of sarcopenic dysphagia, description of care team characteristics, satisfaction of the involved allied health professionals with the programme and difficulties encountered for its implementation.

### Trial design
DYSPHAGING pilot study is a prospective, non-comparative multicentre study conducted in three different geriatric departments and two different hospitals at the university hospital of Lyon (Hospices Civils de Lyon).

### Study sites and participants
The study population will include older patient identified either during their admission (in acute care and rehabilitation units) or during systematic assessments in long-term care units.

Inclusion criteria are: age≥70 years, patient affiliated to a health system, informed of the study (information notice given) and having verbally indicated his/her non-objection to inclusion in the study.

Exclusion criteria are: patient either unable to be fed orally or with an active pathology responsible for acute swallowing disorders (<3 months): neurodegenerative pathology with predominant motor impairment such as Charcot disease, stroke, ear nose and throat pathology, patient under court protection, with progressive somatic or psychiatric pathologies that would impair his/her ability to perform study assessments, or for whom data collection is not possible.

Premature study exit criteria are: refusal to continue the study, transfer to another department within 3 days of screening, death. Data already collected will be kept and analysed.

### Intervention
The DYSPHAGING form was designed as a simple, clear, schematic and pedagogic recto–verso form to be easily handle in routine care (figure 1). The recto face contains the rapid Eating Assessment Tool (EAT-10),[6 7] proposed by the Dysphagia Working Group as one of the most promising screening tools, as it is a self-reported questionnaire, shown to be internally consistent, reproducible and valid.[1] A cut-off score of ≥2 was chosen, as Rofes *et al* demonstrated that it offers 89% sensitivity and 82% specificity for OD.[8] The verso face contains three protection fields: postural manoeuvres, dietary and health rules and adaptation of food textures according to the standardised tool developed by the International Dysphagia Diet Standardization Initiative.[9] The design of the form was developed multidisciplinary with dieticians and a particular attention was paid to the clarity and the understandability of the different schemas.

Following the transmission of an information notice and obtaining an oral consent from patients (and their legal guardian for patients under guardianship) by either a physician or a paramedical professional under his/her responsibility, the intervention involves the integration of patients into a structured screening and care process for sarcopenic dysphagia. The study aims to evaluate the ability of local caregivers, including nursing assistants and nurses in geriatric wards, to adhere to current screening recommendations and implement preventive measures in a routine and standardised manner. Additionally, patient characteristics will be collected at each site through a clinical research assistant (CRA) based on comprehensive medical records. The characteristics of the healthcare team and their satisfaction with the DYSPHAGING form will be assessed during this designated visit.

The intervention process consists of two steps: step 1: recto face of the DYSPHAGING form, consisting of the EAT-10 swallowing disorder screening questionnaire; in case of a score<2, the patient is considered fit for routine care without any additional protection measures; in case of a score≥2, the step 2 should be engaged within 3 days by the healthcare team to implement upper airway

A

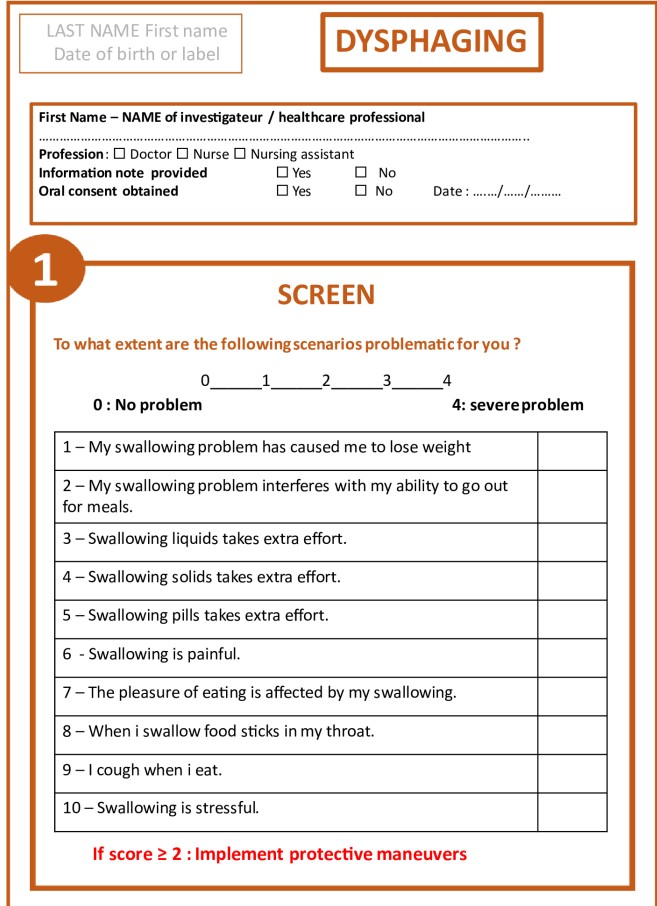

B

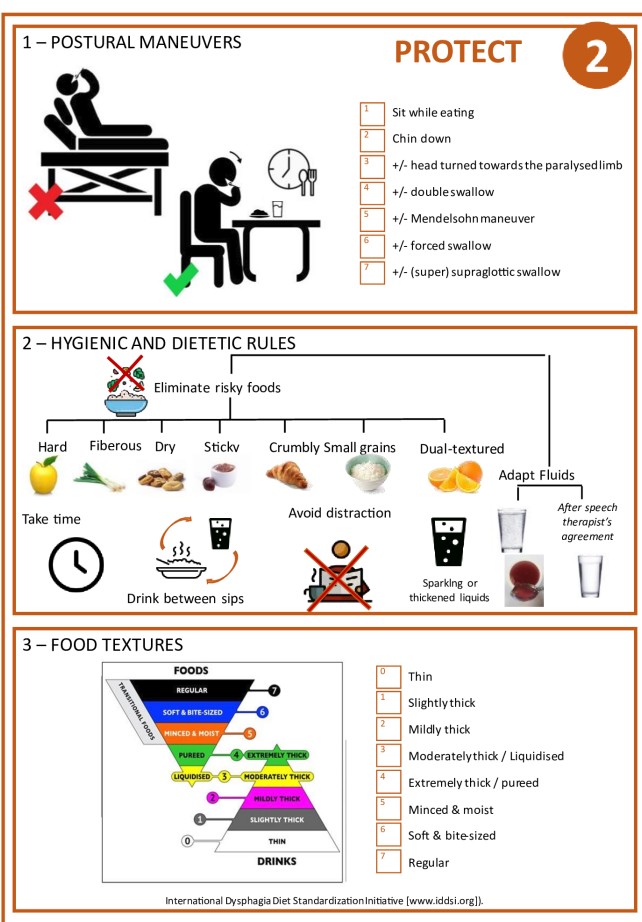

**Figure 1** The DYSPHAGING (dysphagia & aging) form ((A) recto form and (B) verso form).

protection measures within the three protection fields (verso face of the DYSPHAGING form).

Patient characteristics will be collected at each site by a CRA based on comprehensive medical records. During this designated visit, the characteristics of the healthcare team and their satisfaction with the DYSPHAGING educational sheet will be assessed.

### Outcomes and measurements

*The primary outcome* of the study is the proportion of patients who fully complete steps 1 and 2 of the protocol. The endpoint is validated if either:

► Step 1 is completed, and an EAT-10 Score<2.
► Step 1 is completed with an EAT-10 Score≥2 and step 2 is completed within 3 days following step 1.
  *Secondary outcomes of the study* include:
► The percentage of eligible patients who refuse to participate in the study.
► Patient characteristics, such as age, gender, comorbidities, functionality and comedications. Comorbidities will be assessed with the Cumulative Illness Rating Scale—Geriatric; functionality according to the activity of daily living (ADL)[10] and instrumental ADL[11] scores; comedications will be described according to the galenic form and drug class prescribed.

► Description of the factors associated with the risk of sarcopenic dysphagia (malnutrition, defined as either a weight loss >5% in the last 6 months, or >10% beyond 6 months, or a body mass index<22 kg/m²,[12] patient at risk of malnutrition according to the mininutritional assessment short form, neurocognitive disorders, active pulmonary infection, chronic obstructive pulmonary disease, nutritional risk situations).
► The rate of partial completion of the protocol.
► The composition and disciplines of the healthcare team, the level of satisfaction and the difficulties encountered by the involved allied health professionals. A structured questionnaire was specifically designed to evaluate both dimensions (online supplemental document 1). Satisfaction will be explored using Likert scale questionnaires, counting 30 points concerning the initial presentation of the study to the healthcare team, 30 points concerning the feasibility to implement the protection interventions, 30 points concerning difficulties encountered during the study and 2 open questions concerning any missing pieces of information or suggestion to improve the study.

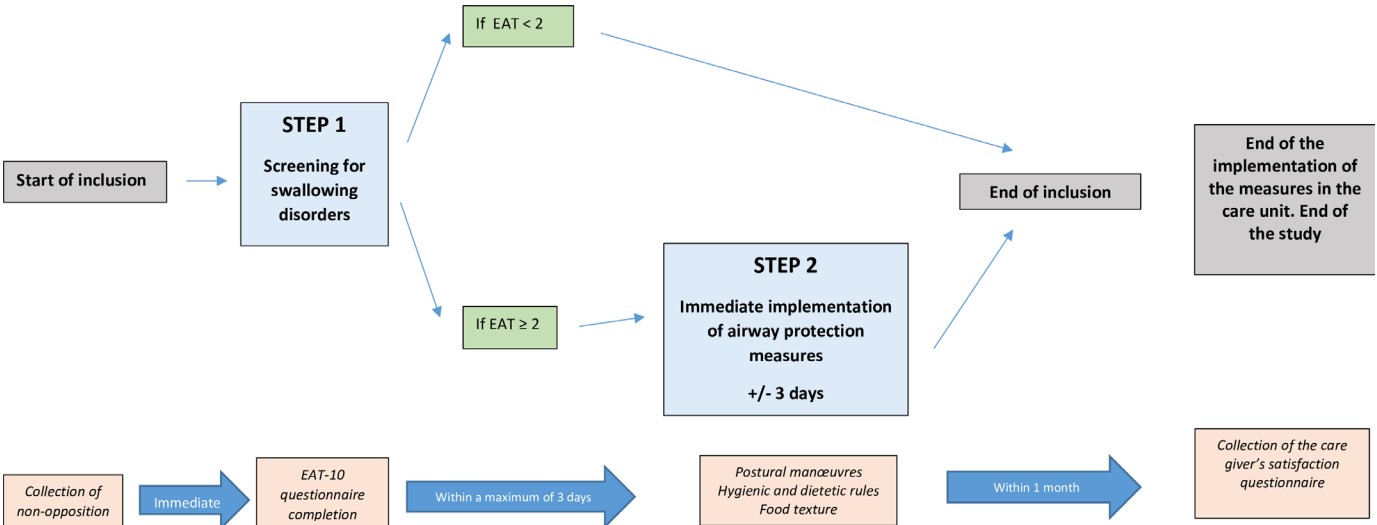

**Figure 2** Design of the DYSPHAGING pilot study. EAT, Eating Assessment Tool.

### Trial conduct

The conduct of the study is represented in figure 2 and table 1.

1. Implementation: training by the principal investigator of the nursing teams at the investigation sites in the materials used in stages 1 and 2 of the DYSPHAGING protocol (EAT-10, checklist of measures to prevent swallowing disorders).
2. Inclusion and screening:
a. Inclusion: information to the patient is provided by either the physician or a paramedical professional under his/her responsibility, collection of non-objection and verification of inclusion and non-inclusion criteria, collection of patient characteristics and clinical data.
b. On the same day as inclusion, performance of step 1 'screening': dispensing of the 10-item EAT-10 screening questionnaire by a paramedical professional.
   1. If EAT-10 Score<2: end of patient participation.
   2. Completion of step 2 if EAT-10 Score≥2: implementation (within 3 days of screening) by the healthcare team of upper airway protection measures appropriate to each patient.

| Table 1 | DYSPHAGING (dysphagia & aging) pilot study: flow diagram | | |
|---|---|---|---|
| **Visits** | **V1** | **V2** | **End of the implementation of the measures** |
| **Time of evaluation** | **Inclusion** | **End of inclusion** | **End of the study** |
| Patient | | | |
| Information notice | X | | |
| Collection of non-opposition | X | | |
| Inclusion and exclusion criteria | X | | |
| Population demographics* | X | | |
| Nutritional risk factors† | X | | |
| Functional independence (ADL, IADL) | X | | |
| Sarcopenic dysphagia risk factors‡ | X | | |
| Sarcopenic dysphagia screening (EAT-10) | X | | |
| Airway protection measures§ | | X | |
| Care team | | | |
| Characteristics of the healthcare staff | | | X |
| Satisfaction questionnaire: Likert scale | | | X |

*Population demographics are age, gender, comorbidities (Cumulative Illness Rating Scale—Geriatric) and comedications.
†Nutritional risk factors are assessed by the Mini Nutritional Assessment.
‡Risk factors for sarcopenic dysphagia include undernutrition, neurocognitive impairment, overt lung infections and chronic obstructive pulmonary disease.
§Upper airway protection recommendations are validated by the following three methods: postural manoeuvres, hygienic–dietary rules, textures within 3 days.
ADL, activity of daily living; EAT-10, Eating Assessment Tool; IADL, instrumental activity of daily living.

Completion of the following checklist:

► Postural manoeuvres (sitting eating, chin down, ±head turned towards the paralysed limb, ±double swallow, ±Mendelsohn manoeuvre, ±forced swallow, ±(super)supraglottic swallow).

► Hygienic and dietary rules (eliminate risky foods, adapt fluids, take time, drink between sips, avoid distraction).

► Food textures (liquid, very slightly thick, slightly thick, moderately smooth/mixed smooth, mixed/pured, ground, swallowing specific soft, normal).
  1. Collection of the satisfaction and difficulties encountered by the involved allied health professionals with the programme (online supplemental table 1).

Strategies for achieving adequate participant enrolment will regularly be implemented using formal (newsletters, posters, meetings) and informal methods to reach target sample size.

### Sample size calculation

The programme will be considered feasible, at the patient level, if the proportion of patients for whom steps 1 and 2 are achievable is statistically higher than 35%, with an anticipated proportion of 50% (= alternative hypothesis). Under these hypotheses, and assuming 10% of patients that might be non-evaluable, the inclusion of 102 patients will be necessary to achieve 90% power to show that the programme is feasible (one-sided alpha risk of 5%). The included patients will be analysed according to the intention-to-treat principle.

### Data management and statistical analyses

A CRA ensures proper study execution, data collection and reporting. Inconsistencies will be reported to the study investigators in order to decide whether the data should be corrected or considered as missing. Adverse health events will be reported to regulatory authorities according to the legislation in force, provided they are aligned with the study's judgement criteria (inhalation/aspiration pneumonia, weight loss, death from any cause). Any changes in the data will be reported. A detailed statistical analysis plan will be drafted before the database is frozen. It will consider any changes in the protocol or unexpected events during the study that have an impact on the analyses presented above. Planned analyses may be completed in line with the study objectives. The analyses will be carried out by an independent statistician with the latest version of the SAS V.9.4 and R (R Core Team. R Foundation for Statistical Computing, Vienna, Austria. URL https:// www.R-project.org/) softwares environment. No intermediate analysis is scheduled.

### Descriptive analyses

A flow diagram will describe the data available for the patient population at baseline and during each follow-up visit. Eligibility criteria for treated patients will be verified, as well as follow-up and end of study visits. Reasons for premature end of study will be provided. Characteristics of the study population, numbers and proportions of missing values will be reported. Patient characteristics will be described using mean and SD or median and IQR for quantitative variables, and frequencies and distribution for categorical variables. A comparison of baseline characteristics between patients with complete follow-up and those with attrition will be performed. Analyses will be performed on the available data, without imputation for missing data.

### Primary analysis

The proportion of patients for whom steps 1 and 2 of the DYSPHAGING form in performed in the 3 days of inclusion will be assessed along with its corresponding 95% CI. Patients for whom information on the completion of steps 1 and 2 is not available will be considered as not having completed these steps.

### Secondary analyses
#### Analyses of the questionnaire for allied health professionals

Analyses will be performed independently using descriptive analyses for quantitative data using mean and SD or median and IQR for Likert scales; overt questions will be reported according to a flat analysis. The analysis of factors associated with sarcopenic dysphagia will be performed by logistic regression. Univariate analyses will be followed by multivariable analyses.

#### Confidentiality

Correspondence tables will be kept in a separate file that does not contain clinical data. The access to the nominative information is protected by a password, and confidentiality is guaranteed by the study.

#### Protocol amendments

A substantial protocol amendment was accepted by the ethics committee on 13 December 2023, to allow the inclusion of patients under guardianship, provided the oral or written consent of their legal guardian. Any important additional modification requiring a new ethics committee approval will be communicated in future publications. Any potential impact of protocol modifications on the results will be discussed as appropriate.

#### Trial status

Patient enrolment began in May 2023. Data are currently being collected.

#### Patient and public involvement

The information letter and consent form for the study were reviewed by a patient partner.

## DISCUSSION
### Discussion of the intervention

Despite growing interest in screening for swallowing disorders, standardised method on which consensus has been reached[1] is not actually implemented in usual care.[5]

The main limitations include the heterogeneity of its presentations, the large number of aetiologies, the poor reproducibility or complexity of screening processes and the need for a clinical confirmation by either a speech specialist or an ear, nose and throat physician. The absence of standardised procedure may lead to disjoined communications between hospital staffs and family carers, leading to suboptimal care, crispation and frustration.[5] In addition, the need for a clinical confirmation of the swallowing problem may postpone the application of prevention procedures.

The aim of the DYSPHAGING approach is to bring together all the care providers around the patient, to ensure a multidisciplinary approach, to use all the time spent with the patient to extract as much relevant information as possible, and to apply as soon as possible, before any clinical confirmation, basic safety measures with the help of a simple and schematic iconography. We believe that the screening and preventive measures proposed by this protocol are appropriate for the healthcare providers working in various geriatric sectors, despite the heterogeneity of the situations encountered in this population. Moreover, the simplicity of the form helps to standardise practices, particularly in a context of high team turnover and may limit the risk of erosion in the application of protection measures, which nevertheless persists. In the future, the DYSPHAGING form is expected to be more widely diffused to caregivers and more generally all care providers, to reach ambulatory care. Due to its simple design, the tool is expected to allow a sharing of upper airway protection measures with the continuum of care providers around the patient, favouring adherence over time.[13]

### Discussion of the trial design

The main aim of this study is to assess the feasibility of screening and various preventive measures. The cut-off value of EAT-10 of 2 was chosen to favour sensitivity over specificity, even if a recent meta-analysis argued for a better diagnostic accuracy with a cut-off value of 3,[14] as the DYSPHAGING form was focused more on screening than diagnosis.[12] It is therefore essential to gather information on the non-implementation of the first steps, to understand the obstacles to the adoption of these initiatives. To simplify the research process and favour adherence by the teams, the primary outcome of the study was intentionally defined as the simplest possible, as the completion of steps 1 and 2 of the protocol, that is, the follow-up ends after 3 days of patients' inclusion. Consequently, the statistical hypothesis did not include any a priori estimation of the rate of patients with an EAT-10 score ≥2 in the studied population, and this information will be of importance in the design of future trials. However, the trial design does not provide any longer-term follow-up of either the maintenance of the protective measures over time or the consequences of oral dysphagia (malnutrition, medical complications, etc), that would have been of interest for exploratory purposes. As healthcare staff are at the centre of diagnosis and care, it is essential to understand the barriers and obstacles they face, by assessing much feedback as possible. Particular attention was paid to the satisfaction of care providers in giving feedback about their training and the work tool. Emphasis was placed on assessing their satisfaction and the ergonomics of the tools made available to them, using a dedicated questionnaire. Future steps in the DYSPHAGING programme of research will have to focus both on the implementation of the DYSPHAGING form in ambulatory care and on satisfaction of the other stakeholders with its ergonomics (patient, caregivers, care providers at home).

The galenic formulation and drug class will also be analysed with care, as iatrogenicity is omnipresent in the geriatric population.

We hope to highlight the various difficulties encountered during this pilot study in order to draw the necessary conclusions for a larger-scale study.

### ETHICS AND DISSEMINATION

The study sponsor is the Hospices Civils de Lyon, responsible for study insurance and pharmacovigilance. The study protocol (V1) was approved by the ethics committee on 15 February 2023; an amended version (V2) was approved on 13 December 2023 and covers all sites involved in this study. The research will be carried out in accordance with the Helsinki Declaration and International Conference on Harmonisation-Good Clinical Practice Guidelines. The trial protocol fulfils the SPIRIT 2013 checklist (online supplemental table 1) and WHO trial registration data set (online supplemental table 2). The study complies with the principles of the data protection act in France and with the General Data Protection Regulation in force in Europe. Each investigator must collect an oral informed consent at the beginning of the procedure. This consent is retained in the patient's medical chart. The patient can stop participation in the study at any time with an oral instruction given to the investigator or CRA. Patients will be informed of additional amendments according to the law in force. The results of the primary and secondary objectives will be published in peer-reviewed journals. All authors of future publications will have to meet the criteria for authorship stated in the Uniform Requirements for Manuscripts Submitted to Biomedical Journals by the International Committee of Medical Journal Editors.

**Author affiliations**
[1]Institut du Vieillissement, Hospices Civils de Lyon, Lyon, Auvergne-Rhône-Alpes, France
[2]Centre de Recherche Clinique Vieillissement, Cerveau, Fragilité, Hôpital des Charpennes, Hospices Civils de Lyon, Villeurbanne, Auvergne-Rhône-Alpes, France
[3]CNRS, UMR5558, Laboratoire de Biométrie et Biologie Evolutive, Lyon, Universite Claude Bernard Lyon 1, Villeurbanne, Auvergne-Rhône-Alpes, France
[4]Service de Biostatistique, Hospices Civils de Lyon, Lyon, Auvergne-Rhône-Alpes, France
[5]Geriatrics, Université Lyon 1 Faculte de Medecine et de Maieutique Lyon-Sud Charles Merieux, Oullins, Rhône-Alpes, France
[6]Hospices Civils de Lyon, Lyon, Auvergne-Rhône-Alpes, France

[7]Unité de Pharmacie clinique oncologique, Hospices Civils de Lyon, Lyon, Auvergne-Rhône-Alpes, France

[8]EA 3738 CICLY, Université Claude Bernard Lyon 1, Villeurbanne, Auvergne-Rhône-Alpes, France

[9]Pôle de Santé Publique, Service Recherche et Epidémiologie cliniques, Hospices Civils de Lyon, 69008 Lyon, France

[10]Research on Healthcare Performance (RESHAPE), Inserm U1290, Université Claude Bernard Lyon 1, 69008 Lyon, Auvergne-Rhône-Alpes, France

[11]Direction à la Recherche en Santé, Hospices Civils de Lyon, Lyon, Auvergne-Rhône-Alpes, France

[12]Plateforme Transversale de Recherche de l'ICHCL, Hospices Civils de Lyon, Lyon, Auvergne-Rhône-Alpes, France

[13]Service de Gériatrie, Groupement Hospitalier Sud, Hospices Civils de Lyon, Lyon, Auvergne-Rhône-Alpes, France

[14]Service de Gériatrie, Centre Hospitalier de la Croix-Rousse, Hospices Civils de Lyon, Lyon, France

[15]CarMeN Laboratory, Inserm U1060, INRA U1397, Université Claude Bernard Lyon 1, INSA Lyon, Charles Mérieux Medical School, University of Lyon, Oullins, France

**Acknowledgements** The authors acknowledge the teams of Lyon Sud Hospital, Pierre Garraud Hospital who contribute to patient enrolment in this study. The authors would like to thank the Centre de Recherche Clinique (Clinical Research Center) Vieillissement Cerveau Fragilité and the Direction à la Recherche en Santé (Health Research Department) of the Hospices Civils de Lyon for their valuable help in trial design and conduct.

**Contributors** OD, ST-d-M, NM-D, FS, ZN-S, CH, LG, KG, SM, MC, MM and CF participated to the trial design conception. KG, ST-d-M, AS, DD and CF managed fundraising and grant follow-up. OD led the drafting of the manuscript. All authors critically reviewed and approved the final version of the protocol.

**Funding** This work was supported by the Institut Nutrition (Prix de l'Institut Nutrition 2021) and the Fondation de l'Avenir (Grant No. MLHR2023-89).

**Competing interests** None declared.

**Patient and public involvement** Patients and/or the public were involved in the design, or conduct, or reporting, or dissemination plans of this research. Refer to the Methods and analysis section for further details.

**Patient consent for publication** Not applicable.

**Provenance and peer review** Not commissioned; externally peer reviewed.

**ORCID iD**
Claire Falandry http://orcid.org/0000-0001-7267-4723

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
