## [Reviewer comments · BMJ Open]

ARTICLE DETAILS

TITLE (PROVISIONAL)	Feasibility of a screening and prevention procedure for risks associated with dysphagia in older patients in geriatric units: the DYSPHAGING pilot study protocol
AUTHORS	Durlach, Olivier; Tripoz-dit-Masson, Stéphanie; Massé-Deragon, Nicolas; SUBTIL, Fabien; Niasse-Sy, Zeinabou; HERLEDAN, Chloé; Guittard, Laure; Goldet, Karine; Merazga, Salima; Chabert, Margaux; Suel, Anne; Dayde, David; Merdinian, Marion; Falandry, Claire

VERSION 1 – REVIEW

REVIEWER	Anantapong, Kanthee University College London, Marie Curie Palliative Care Research Department
REVIEW RETURNED	09-Dec-2023

GENERAL COMMENTS	Thank you for this interesting protocol. It aims to pilot the intervention to help identify and support older people with dysphagia. The research seems to be important and would provide useful information. The protocol is quite well-written; however, it can be improved by considering the following feedback and suggestions to make it more interesting, clear and justified. 1. Strengths and limitations (page 6): The current strengths and limitations seem to be general information about the intervention. Could the authors highlight the differences between this intervention and existing interventions and the expected benefits from this intervention? The limitations of this intervention are also not mentioned in this section.2. Introduction, paragraph 1: In older adults, cognitive declines and psychological and behavioural changes may impact their ability to eat and drink, and impaired communication in this population makes the decisions and care for nutrition and hydration even more difficult. The use of the tool in this current study may facilitate such a process. The authors may wish to address this point and include the paper by Anantapong et al. (2022) (https://doi.org/10.1093/ageing/afac230).3. Intervention (paragraph 2 page 9): How do the researchers identify potential participants? Who will be the first person to approach the eligible or potential participants? With the transmission of an information notice, will it be self-identification or self-enrollment to the study?4. Intervention (paragraph 2 page 9): Who will provide the tools to the patients? Will they be trained before doing this, and how to do so? Will their caregivers be involved in the consent process? Will cognitive assessment be performed before recruitment or data collection?
---

	5. Intervention (page 9, paragraph 2, line 42): Will the recto face be used with the help of the research team and/or family members or self-administered by the participants? 6. Intervention (page 9, paragraph 2, line 49, within 3 days by the healthcare team): Will the patients and their family carers acknowledge or read the information in verso face? How will they be involved in such measures/care provided in the verso face? 7. Intervention: How long step 2 will be provided to the patients? When will the outcomes be assessed - how often and how many times? 8. Intervention (page 11, paragraph 1): Will the satisfaction also be assessed by patients and their family caregivers? Why not do so? This may include limitations, recommendations, or plans for future research. 9. Intervention (outcomes and measurements): Will the researchers assess clinical or patient outcomes? What are they? This may include limitations, recommendations, or plans for future research. 10. Step 2, measures/intervention: Will the family carers help with these strategies? Will the intervention be used for inpatients or institutionalised older adults? Have the authors planned for the sustainability of the intervention? Can it apply to a community setting? This may include limitations, recommendations, or plans for future research.
--	---

REVIEWER	Fernandez-Araque, Ana Universidad de Valladolid
REVIEW RETURNED	11-Dec-2023

GENERAL COMMENTS	The topic is important for nursing and its care and prevention. Would you do any minor review on how they have calculated, and based on which sample size? How have you taken into account the application of this pilot study among the different areas where professionals have participated? They say it is multicentric, but it has only been done in 3 rooms of a hospital, so it is not multicentric. The quality of the study is greatly impoverished if it is not shared.
--

VERSION 1 – AUTHOR RESPONSE

Reviewer 1 - Dr. Kanthee Anantapong, University College London, Prince of Songkla University
Faculty of Medicine

Thank you for this interesting protocol. It aims to pilot the intervention to help identify and support older people with

dysphagia. The research seems to be important and would provide useful information.

Reviewer 1, General comment: General overview

The protocol is quite well-written; however, it can be improved by considering the following feedback and suggestions

to make it more interesting, clear and justified.

Author's response: We thank reviewer 1 for the good overall appreciation of the manuscript, and we appreciate the

constructive comments to make it clearer and more justified.

Reviewer 1, Comment 1: Strengths and limitations (page 6)

The current strengths and limitations seem to be general information about the intervention. Could the authors

highlight the differences between this intervention and existing interventions and the expected benefits from this

intervention? The limitations of this intervention are also not mentioned in this section.

Authors' response: The manuscript was edited as follows,

- To highlight the differences between this intervention and existing interventions (Line 295):

Discussion of the intervention

Despite growing interest in screening for swallowing disorders, there is no standardized method on which consensus

has been reached (1) are not actually implemented in usual care (5). Among the The main limitations include the

heterogeneity of its presentations, the large number of etiologies, the poor reproducibility or complexity of screening

processes and the need for a clinical confirmation by either a speech specialist or an ear, nose and throat physician. The

absence of standardized procedure may lead to disjointed communications between hospital staffs and family carers,

leading to suboptimal care, crisparation and frustration (5). In addition, the need for a clinical confirmation of the

swallowing problem may postpone the application of prevention procedures.

- To highlight the benefits of the intervention (Lines 306-319):

“The aim of the DYSPHAGING approach is to bring together all the care providers around the patient healthcare

professionals involved in the patient's care, to ensure a multi-disciplinary approach, and to use all the time spent with

the patient to extract as much relevant information as possible, and to apply as soon as possible, before any clinical

confirmation, basic safety measures with the help of a simple and schematic iconography. We believe that the screening

and preventive measures proposed by this protocol are appropriate for the healthcare providers working in various

geriatric sectors, despite the heterogeneity of the situations encountered in this population. Moreover, the simplicity of

the form helps to standardize practices, particularly in a context of high team turnover and may limit the risk of erosion

in the application of protection measures, which nevertheless persists. In the future, the DYSPHAGING form is expected

to be more widely diffused to caregivers and more generally all care providers, to reach ambulatory care. Due to its

simple design, the tool is expected to allow a sharing of upper airway protection measures with the continuum of care

providers around the patient, favoring adherence (13).

3

- To highlight the limitations of the intervention:

Considering the (unexplored) risk of a reduced alertness and application of the preventive measures over time

- Line 310:

(...) We believe that the screening and preventive measures proposed by this protocol are appropriate for the healthcare

providers working in various geriatric sectors, despite the heterogeneity of the situations encountered in this population.

Moreover, the simplicity of the form helps to standardize practices, particularly in a context of high team turnover and may limit the risk of erosion in the application of protection measures, which nevertheless persists. In the future, the DYSPHAGING form is expected to be more widely diffused to caregivers and more generally all care providers, to reach ambulatory care. Due to its simple design, the tool is expected to allow a sharing of upper airway protection measures with the continuum of care providers around the patient, favoring adherence over time (13).

- Line 332:

“However, the trial design does not provide any longer term follow up of either the maintenance of the protective measures over time or the consequences of oral dysphagia (malnutrition, medical complications, etc), that would have been of interest for exploratory purposes.”

Reviewer 1, Comment 2: Introduction, paragraph 1

In older adults, cognitive declines and psychological and behavioural changes may impact their ability to eat and drink, and impaired communication in this population makes the decisions and care for nutrition and hydration even more difficult. The use of the tool in this current study may facilitate such a process. The authors may wish to address this

point and include the paper by Anantapong et al. (2022) (<https://doi.org/10.1093/ageing/afac230>).

Author’s response: We appreciate this comment. We expect that the standardization among team members of the

questioning of swallowing problems using a systematic screening – avoiding any risk of subjectivity or misinterpretation

by the patients and their families, may attenuate frustration and crispation and, on the contrary, may give the

opportunity to talk about eating and drinking difficulties and anticipate future problems. The

introduction was

amended as follows

- Line 89 (Introduction section):

“Although recent awareness of the high prevalence of sarcopenic dysphagia and its severe consequences among older

individuals with disabilities and hospitalized patients has grown, the screening within the affected population remains

low and challenging, leading to suboptimal care (5).”

- Line 301 (Discussion section):

The absence of standardized procedure may lead to disjointed communications between hospital staffs and family carers,

leading to suboptimal care, crispation and frustration (5).

Reviewer 1, Comment 3: Intervention (paragraph 2 page 9)

How do the researchers identify potential participants? Who will be the first person to approach the eligible or potential

participants? With the transmission of an information notice, will it be self-identification or self-enrollment to the

study?

4

Author’s response: A certain degree of liberty/autonomy is left to the different units/wards:

- Considering the identification of the participants: in some of them, when the patients’ arrivals can be

anticipated, their clinical charts may be analyzed for pre-screening on inclusion and exclusion criteria. For

others, the identification is left to the member of the team who will welcome the patient.

- Considering the first person to approach eligible patients: in some unit/wards, a nurse or a nurse assistant is

dedicated to welcoming new patients and will be the first to present the protocol to them; in some others, the

geriatrician (physician) is performing a clinical exam before letting the paramedics perform their initial workup.

These details have been inserted in the manuscript as follows (Line 154):

“Following the transmission of an information notice and obtaining an oral consent from patients by either a physician

or a paramedical professional under his/her responsibility, the intervention involves the integration of patients into a

structured screening and care process for sarcopenic dysphagia.”

Reviewer 1, Comment 4: Intervention (paragraph 2 page 9)

Who will provide the tools to the patients? Will they be trained before doing this, and how to do so?

Will their caregivers

be involved in the consent process? Will cognitive assessment be performed before recruitment or data collection?

Author’s response: The Dysphaging form is expected to be provided by the first-line professional, under the

responsibility of the physician. In case of any doubt considering the cognitive status of the patient, this professional can

request a verification by the physician of the inclusion/exclusion criteria. It should be noticed that a substantial

amendment to the protocol was validated by the ethics committee on the December, to respond to a frequently

encountered difficulty concerning patients under guardianship, who were previously excluded, leading to the exclusion

of a significant proportion of patients in long-term care units and raising ethical questions, as these patients can also

benefit from a dysphagia screening protocol. Since this amendment, inclusion has been possible, provided the guardian

gives verbal or written consent. Considering the role of caregivers, outside the particular role of legal guardians, it is

encouraged, since our ambition is to spread the Dysphaging form to ambulatory care in the future, and in this

perspective, caregivers will be essential relays in the home. However, this pilot study focuses on the feasibility of the

program by hospital teams, and the role of the caregiver in that particular study is limited to patient support and advice

in compliance with declaration of Helsinki and current French regulatory rules.

These details have been inserted in the manuscript as follows:

- Line 154:

“Following the transmission of an information notice and obtaining an oral consent from patients (and their guardian

for patients under guardianship) by either a physician or a paramedical professional under his/her responsibility, the

intervention involves the integration of patients into a structured screening and care process for sarcopenic dysphagia.”

- Lines 202-210

“Trial conduct

(...)

2) Inclusion and screening

5

a) Inclusion: Information to the patient is provided by either the physician or a paramedical professional under his/her responsibility, collection of non-objection and verification of inclusion and non-inclusion criteria, collection of patient characteristics and clinical data.”

- Line 281:

“Protocol amendments

A substantial protocol amendment was accepted by the ethics committee on December 13th, 2023, to allow the inclusion

of patients under guardianship, provided the oral or written consent of their legal guardian. Any important additional

modification requiring a new ethics committee approval will be communicated in future publications.

Any potential

impact of protocol modifications on the results will be discussed as appropriate. ”

- Lines 295-306

« Discussion of the intervention

(...)

We believe that the screening and preventive measures proposed by this protocol are appropriate for the healthcare

providers working in various geriatric sectors, despite the heterogeneity of the situations encountered in this

population. Moreover, the simplicity of the form helps to standardize practices, particularly in a context of high team

turnover and may limit the risk of erosion in the application of protection measures, which

nevertheless persists. In the

future, the DYSPHAGING form is expected to be more widely diffused to caregivers and more

generally all care

providers, to reach ambulatory care.”

- Line 349-353

“Ethics and dissemination

The study sponsor is the Hospices Civils de Lyon, responsible for study insurance and pharmacovigilance. The study

protocol (V1) was approved by the ethics committee on February 15, 2023; an amended version (V2) was approved on

December 13, 2023 and covers all sites involved in this study. “

Reviewer 1, Comment 5: Intervention (page 9, paragraph 2, line 42):

Will the recto face be used with the help of the research team and/or family members or self-administered by the participants?

Author’s response: The whole procedure is expected to be fulfilled by the paramedical members under the

responsibility of the physician; the EAT10 is proposed as a hetero-questionnaire, to induce clinical reflexes and allow

the rapid implementation of protective maneuvers (verso face of the form).

This point has been clarified as follows:

- Lines 202-212

“Trial conduct

(...)

2) Inclusion and screening

(...)

6

b) On the same day as inclusion, performance of step 1 "Screening": dispensing of the 10-item EAT-10 screening

questionnaire by a paramedical professional

Reviewer 1, Comment 6: Intervention (page 9, paragraph 2, line 49, within 3 days by the healthcare team):

Will the patients and their family carers acknowledge or read the information in verso face? How will they be involved

in such measures/care provided in the verso face?

Author's response: We thank reviewer 1 for this important comment, as the purpose of the whole DYSPHAGING

program is to increase the overall awareness of patients, paramedical teams and all stakeholders to the problem of

dysphagia. However, and as previously clarified, the main focus of this pilot study is to evaluate the reception of the

tool by the hospital teams. Future work will explore its reception by patients, caregivers and care providers at home.

Even if the implication of the caregiver is not specifically explored, there is an extensive piece of data highlighting the

positive impact that sharing medical information with the continuum of care providers around the patient, and in

particular with the caregiver, has on patients' adherence (1).

This point has been emphasized in the discussion as follows:

- Lines 295-319

« Discussion of the intervention

(...)

We believe that the screening and preventive measures proposed by this protocol are appropriate for the healthcare

providers working in various geriatric sectors, despite the heterogeneity of the situations encountered in this population.

Moreover, the simplicity of the form helps to standardize practices, particularly in a context of high team turnover and

may limit the risk of erosion in the application of protection measures, which nevertheless persists. In the future, the

DYSPHAGING form is expected to be more widely diffused to caregivers and more generally all care providers, to reach

ambulatory care. Due to its simple design, the tool is expected to allow a sharing of upper airway protection measures

with the continuum of care providers around the patient, favoring adherence over time (13).

Reviewer 1, Comment 7: Intervention:

How long step 2 will be provided to the patients? When will the outcomes be assessed - how often and how many

times?

Author's response: The upper airway protection measures are expected to be implemented in the long-term, during

the current hospitalization of the patient and even in successive environments (at home, in a rehabilitation unit, etc).

However, this specific point is not explored due to the trial design. According to it, the patient's follow-up ends on day

3 after inclusion. The choice of the primary outcome (Line 175):

“The primary outcome of the study is the proportion of patients who fully complete steps 1 and 2 of the protocol. The endpoint is validated if either:

- Step 1 is completed, and an EAT-10 score < 2, or
- Step 1 is completed with an EAT-10 score ≥ 2 and step 2 is completed within 3 days following step 1.”

7

May be debated and has been discussed as follows (Line 325-335):

“It is therefore essential to gather information on the non-implementation of the first steps, to understand the obstacles to the adoption of these initiatives. To simplify the research process and favor adherence by the teams, the primary outcome of the study was intentionally defined as the simplest possible, as the completion of steps 1 and 2 of the protocol, ie the follow-up ends after 3 days of patients' inclusion. Consequently, the statistical hypothesis did not include any a priori estimation of the rate of patients with an EAT10 score ≥ 2 in the studied population, and this information will be of importance in the design of future trials. However, the trial design does not provide any longer term follow up of either the maintenance of the protective measures over time or the consequences of oral dysphagia (malnutrition, medical complications, etc), that would have been of interest for exploratory purposes.”

Reviewer 1, Comment 8: Intervention (page 11, paragraph 1):

Will the satisfaction also be assessed by patients and their family caregivers? Why not do so? This may include limitations, recommendations, or plans for future research.

Author's response: We agree with this point, which should be questioned in future DYSPHAGING implementation studies (Lines 338-342):

“Emphasis was placed on assessing their satisfaction and the ergonomics of the tools made available to them, using a dedicated questionnaire. Future steps in the DYSPHAGING program of research will have to focus both on the implementation of the DYSPHAGING form in ambulatory care and on satisfaction of the other stakeholders with its ergonomics (patient, caregivers, care providers at home). “

Reviewer 1, Comment 9: Intervention (outcomes and measurements):

Will the researchers assess clinical or patient outcomes? What are they? This may include limitations, recommendations, or plans for future research.

Author's response: As previously mentioned in response to Comment 7, the primary outcome was intentionally the simplest possible, leading to end the patient's participation 3 days after its inclusion. Consequently, the trial design excludes any longer follow up; this point has been added in the discussion as follows (Lines 325-335): “It is therefore essential to gather information on the non-implementation of the first steps, to understand the obstacles to the adoption of these initiatives. To simplify the research process and favor adherence by the teams, the primary outcome of the study was intentionally defined as the simplest possible, as the completion of steps 1 and 2 of the

protocol, ie the follow-up ends after 3 days of patients' inclusion. Consequently, the statistical hypothesis did not include any a priori estimation of the rate of patients with an EAT10 ≥ 2 in the studied population, and this information will be of importance in the design of future trials. However, the trial design does not provide any longer term follow up of either the maintenance of the protective measures or the consequences of oral dysphagia (malnutrition, medical complications, etc), that would have been of interest for exploratory purposes."

Reviewer 1, Comment 10: measures/intervention:

8

Will the family carers help with these strategies? Will the intervention be used for inpatients or institutionalised older

adults? Have the authors planned for the sustainability of the intervention? Can it apply to a community setting? This

may include limitations, recommendations, or plans for future research.

Author's response: As raised in response to Comment 6, this remark is perfectly right and was added in the discussion

as follows (Line 295-319):

« Discussion of the intervention

(...)

We believe that the screening and preventive measures proposed by this protocol are appropriate for the various

geriatric sectors, despite the heterogeneity of the situations encountered in this population. Moreover, the simplicity of

the form helps to standardize practices, particularly in a context of high team turnover and may limit the risk of erosion

in the application of protection measures, which nevertheless persists. In the future, the

DYSPHAGING form is expected

to be more widely diffused to caregivers and more generally all care providers, to reach ambulatory care. Due to its

simple design, the tool is expected to allow a sharing of upper airway protection measures with the continuum of care

providers around the patient, favoring adherence over time (13)."

Reviewer 2: Dr. Ana Fernandez-Araque, Universidad de Valladolid

Reviewer 2, Comment 1: General overview

The topic is important for nursing and its care and prevention.

Author's response: We thank reviewer 2 for highlighting the importance of the topic.

Reviewer 2, Comment 2

Would you do any minor review on how they have calculated, and based on which sample size?

Author's response: The statistical hypothesis of the trial is not based on any proposed estimation of the rate of EAT10

score ≥ 2 in the different clinical contexts (acute care unit, rehabilitation unit, long-term care unit) but on the ability of

the teams to validate the procedure in the 3 days after the patient's inclusion, ie:

- If EAT10 is < 2 , no specific intervention.

- If EAT10 is ≥ 2 , the implementation of the prevention procedures in the 3 days after patient's inclusion

A rate of 50% is anticipated (alternative hypothesis); to exclude the null hypothesis of a rate $\leq 35\%$ the number of

patients needed is 92 (unilateral test, $\alpha = 5\%$, power $(1-\beta) = 90\%$, epiR package 0.9-96); considering a 10% rate of lost

of follow-up, the total number of patients to be included was fixed at 102.

The choice of the primary endpoint, and consequently of the statistical hypothesis was discussed in the Discussion

section as follows (Lines 321-332):

“Discussion of the trial design

The main aim of this study is to assess the feasibility of screening and various preventive measures.

The cutoff value of

EAT10 of 2 was chosen to favor sensitivity over specificity, even if a recent meta-analysis argued for a better diagnostic

9

accuracy with a cutoff value of 3, as the DYSPHAGING form was focused more on screening than diagnosis (12). It is

therefore essential to gather information on the non-implementation of the first steps, to understand the obstacles to

the adoption of these initiatives. To simplify the research process and favor adherence by the teams, the primary

outcome of the study was intentionally defined as the simplest possible, as the completion of steps 1 and 2 of the

protocol, ie the follow-up ends after 3 days of patients' inclusion. Consequently, the statistical hypothesis did not include

any a priori estimation of the rate of patients with an EAT10 score ≥ 2 in the studied population, and this information will

be of importance in the design of future trials.”

Reviewer 2, Comment 3

How have you taken into account the application of this pilot study among the different areas where professionals have

participated? They say it is multicentric, but it has only been done in 3 rooms of a hospital, so it is not multicentric.

The quality of the study is greatly impoverished if it is not shared.

Author's response:

The Hospices Civils de Lyon gather 24 000 professionals in 13 different hospitals, present in very different cities in the

area of Lyon (<https://www.chu-lyon.fr/hospices-civils-de-lyon>). The Dysphaging pilot study was implemented in 2 of

these hospitals, and in 3 different departments headed by different managers, with different care orientations (acute

care unit at Lyon Sud Hospital, Pierre Bénite city; rehabilitation units and long-term care units at Pierre Garraud

Hospital, Lyon city). All three sites are not used to participate to clinical trials, requiring a very high involvement of CRA

for the protocol implementation. We agree with the reviewer's comment that increasing the number of centers will be

requested in the future, to validate the generalization of the tool; however, the first months of implementation of the

trial revealed many resistances from the team members that we will have to overcome before such generalization.

The manuscript was edited as follows, to clarify these points (Line 123):

“Trial design

DYSPHAGING pilot study is a prospective, non-comparative multicentre study conducted in three different geriatric

departments and two different hospitals at the university hospital of Lyon (Hospices Civils de Lyon).“

Editors' Comments to Author

Please ensure that your protocol reports all outcome measures for your trial and ensure that the primary and secondary outcome measures are consistent between your protocol article and the trial registry.

Author's response:

The ClinicalTrials registry was updated with the substantial modification amendment validated on December 13th, 2023:

<https://classic.clinicaltrials.gov/ct2/show/NCT05734586>

VERSION 2 – REVIEW

REVIEWER	Anantapong, Kanthee University College London, Marie Curie Palliative Care Research Department
REVIEW RETURNED	27-Feb-2024
GENERAL COMMENTS	Thank you for the revisions. The authors have successfully addressed all the comments. I have no further comments. Congratulations!